# The Benefits of Physical Activity for People with Obesity, Independent of Weight Loss: A Systematic Review

**DOI:** 10.3390/ijerph19094981

**Published:** 2022-04-20

**Authors:** Rachele Pojednic, Emma D’Arpino, Ian Halliday, Amy Bantham

**Affiliations:** 1Department of Health and Human Performance, Norwich University, Northfield, VT 05663, USA; 2Institute of Lifestyle Medicine, Harvard Medical School, Boston, MA 02115, USA; 3Harvard Extension School, Cambridge, MA 02138, USA; emma.darpino@gmail.com (E.D.); iah963@g.harvard.edu (I.H.); 4MGH Institute of Health Professions, Charlestown, Boston, MA 02129, USA; 5Move to Live More, LLC, Somerville, MA 02144, USA; abantham@movetolivemore.com

**Keywords:** obesity, physical activity, weight loss, health

## Abstract

Purposeful weight loss continues to be the primary focus for treating obesity. However, this strategy appears to be inadequate as obesity rates continue to rise and a myriad of benefits of physical activity that affect multiple health outcomes related to obesity and associated comorbidities are not integrated into treatment strategies. There are emerging correlational data in individuals with obesity that demonstrate physical activity can be beneficial to many critical health markers, independent of weight loss or changes in BMI. This systematic review investigates interventional studies that examine health markers, independent of weight loss, in individuals with obesity. Fourteen studies were identified that utilized a variety of physical activity interventions with primary endpoints that included cellular, metabolic, systemic and brain health outcomes. The review of the literature demonstrates that for individuals with obesity, there are both small-scale and large-scale physiologic benefits that occur with increased physical activity of various modalities. Focusing on these benefits, rather than a narrow focus of weight loss alone, may increase physical activity behavior and health for individuals with obesity.

## 1. Introduction

Purposeful weight loss continues to be the primary focus for treating obesity. Behavioral recommendations for weight loss include lifestyle modifications that reduce caloric intake from diet and increase caloric output with increased physical activity [1,2]. However, this strategy appears to be inadequate as obesity rates continue to rise and the myriad of benefits of physical activity that affect multiple health outcomes related to obesity and associated comorbidities are not integrated into treatment strategies.

Independent of weight loss or changes in body mass index (BMI), there are emerging correlational data in individuals with obesity that demonstrate physical activity can be beneficial to many critical health markers [3]. Epidemiologic studies noted specific relationships between physical activity and health outcomes that include increased cardiorespiratory and muscle fitness [4,5] as well as decreased risk of all-cause mortality and cardiovascular disease [6]. In a prospective epidemiological study, metabolically healthy but obese individuals had a lower risk of all-cause mortality, non-fatal and fatal cardiovascular disease and cancer mortality than their metabolically unhealthy obese peers, while no significant differences were observed between metabolically healthy but obese and metabolically healthy normal-fat individuals [7]. Furthermore, data indicate that physical activity can improve other markers of health in normal-weight and overweight individuals, such as overall quality of life [8], brain health [9], cognition [10], memory [11], sleep [12] and anxiety [13]. Yet, there are limited intervention studies specifically examining individuals with obesity that investigate health markers independent of weight loss as primary outcomes [3]. This is a major gap in the literature, and there have been recent calls to characterize outcomes in people with obesity, particularly in obesity subtypes [14].

By not investigating the benefits of physical activity independent of weight loss, there is a missed opportunity to enhance the physical and mental health of individuals with obesity, perhaps even leading to incorrect courses of treatment by exclusively recommending pounds of weight lost as the primary outcome [3]. This systematic review was designed to examine interventions with outcomes related to the effects of physical activity on health markers, independent of weight loss, in individuals with obesity. The intention was to highlight potential gaps in the literature and to understand which health outcomes are causally affected by physical activity in this understudied population. Based upon the evidence available, we identified four specific physical activity outcomes that affect individuals with obesity, independent of weight loss and include cellular, metabolic (i.e., fuel utilization), systemic (i.e., cardiovascular) and brain health outcomes.

## 2. Materials and Methods

A systematic review method that complied with the Preferred Reporting Items for Systematic Review and Meta-Analysis Protocols (PRISMA-P) was used [15]. The search engines Pubmed, Embase and PsychInfo were primarily used to yield relevant studies regarding the benefits of physical activity, independent of weight loss, for adults with obesity (BMI > 30 kg/m^2^). The following terms were included in all database searches: obese *, physical activity, fitness and exercise. The terms found in Table 1 were used to refine each search.

Inclusion criteria consisted of articles originally published in English between the years 2011 and 2021 with average participant BMI reported over 30 kg/m^2^. Studies that included participants with an average BMI < 30 kg/m^2^ were excluded. Studies that included BMI change or weight loss as the main outcome were excluded, along with studies that did not control for weight loss. Studies with dietary and/or supplemental interventions were also excluded.

In order to increase inter-rater reliability, upon completion of abstract review, all four members of the research team assessed remaining manuscripts for final eligibility. This included ensuring that the study design included a physical activity intervention, and that weight loss was not a primary end point in the study. The studies were also screened to ensure that the inclusion criteria of BMI of the study participants was >30 kg/m^2^ as to only include people specifically with obesity. Studies remained included if the design included a priori intent to recruit subjects with a BMI > 30 kg/m^2^ to compare with a normal BMI cohort (Figure 1). Studies were not excluded for any noted limitations by the authors.

## 3. Results

### 3.1. Included Articles

There were 14 studies that were included in the final review (Table 2). The included studies examined a wide range of physical activity interventions that incorporated sports, aerobic and resistance training combinations, sprint cycling, walking, high intensity interval training and moderate continuous training that were either individually monitored or overseen by a trainer. Based on primary outcomes, studies were divided for analysis into four major categories of the effects of the various physical activity modalities: cellular, metabolic and cardiovascular, systemic and brain health.

### 3.2. Cellular Outcomes

Three studies reported that physical activity, independent of weight loss, resulted in positive changes of cellular biomarkers in people with obesity, including increases in telomere length [17], improvement in anti-oxidative and anti-thrombotic enzymes associated with HDL cholesterol [29] and skeletal muscle microvascular enzymes that affect capillary density and vasoreactivity [18]. Physical activity interventions in these studies included combined aerobic and resistance training protocols [17,29], as well as sprint or moderate continuous cycling [18].

Brandao and colleagues [17] reported that physical activity increased telomere length in people with obesity regardless of weight loss. Telomeres are protective protein complexes at the end of chromosomes, which limit the chromosome shortening caused by replication. Telomere shortening has been correlated with metabolic disorders and telomere maintenance and lengthening with longevity. In this study, a convenience sample of 13 pre-menopausal women who had a BMI of 30–40 kg/m^2^ underwent an 8-week physical activity intervention program that included 55 min of combined aerobic exercise and resistance training three times per week. Upon completion of the training program, the researchers found no change to body weight or BMI (*p* < 0.05), but a 6% increase in fat free mass (kg) (6%) (*p* < 0.05) and 8% increase in VO2 max (ml/kg/min) (*p* < 0.05), as well as a 2% decrease in weight circumference (cm) (*p* < 0.05). Researchers also found a significant increase in telomere length (1.03 ± 0.04 to 1.07 ± 0.04 T/S ratio *p* = 0.001). Telomere length, which is denoted as T/S ratio, was determined by telomere to single gene copy ratio ΔCt (Ct(telomeres)/Ct(single-gene)). Although this paper focused on effects independent of weight loss, this study also identified an inverse relationship between telomere length and waist circumference before (r = −0.536, *p* = 0.017, r^2^ = 0.117) and after the exercise regimen (r = −0.655 *p* = 0.015, r^2^ = 0.321). Limitations of this study included recruitment by convenience and lack of a control group.

Wouldberg and colleagues [29] reported the role of exercise on lipid profile, HDL functionality and inflammation in 32 South African women with obesity, who were block randomized to either exercise intervention or no change groups. The exercise intervention consisted of 12 weeks of aerobic and resistance exercise training of moderate-vigorous intensity (75–80% peak heart rate) for 40 to 60 min, 4 days per week supervised by a trained exercise physiologist. Cardiovascular exercises included aerobic dance, running, skipping and stepping. Resistance training included body weight and equipment-based exercises (e.g., bands and free weights) that incorporated squats, lunges, bicep curls, push-ups and shoulder press with a prescribed intensity of 60% to 70% heart rate peak. The lipid profile and HDL functionality were measured by evaluating the levels of cellular cholesterol efflux capacity, reduction in endothelial vascular cell adhesion expression and paraxonase (PON) and platelet activating factor acetylhydrolase (PAF-AH). PON and PAF-AH are enzymes that are associated with HDL and have anti-oxidative and anti-thrombotic effects, respectively. Statistical models were corrected for changes in BMI where appropriate. At the end of the study, the researchers reported significant decreases in the exercise group while the control group had significant increases for the following outcomes: BMI (−1.0 ± 0.5% vs. + 1.2 ± 0.6%, *p* = 0.010), PON activity (−8.7 +/− 2.4%, +1.1 +/− 3.0%, *p* = 0.021), PAF-AH serum expression (−22.1 ± 8.0% vs. + 16.9 ± 9.8, *p* = 0.002) and distribution of small HDL subclasses(−10.1 ± 5.4% vs. + 15.7 ± 6.6%, *p* = 0.004). Limitations of this study include a small sample size overall with only a subset of subjects undergoing some biological assays.

Cocks et al. [18] aimed to determine the effects that 4 weeks of constant workload through either sprint interval training (SIT) or moderate intensity continuous training (MICT) would have on skeletal muscle microvascular density and microvascular filtration capacity in previously sedentary young men with obesity. Additionally, they evaluated the effects these training programs would have on skeletal muscle microvascular enzymes (NADPH oxidase 2 and endothelial nitric oxide synthase) responsible for nitric oxide bioavailability as well as arterial stiffness and blood pressure. A total of 16 men (age 25 ± 1 years), with obesity were randomly assigned to either SIT or MICT groups, in a matched fashion based on age, BMI and VO2 peak. Groups participated in 4 weeks of MICT (40–60 min cycling at 65% VO2 peak, 5 times per week) or constant load SIT (4–7 constant workload intervals of 200% Wmax 3 times per week). Results demonstrated SIT and MICT have equal benefits on aerobic capacity, insulin sensitivity, muscle capillarization and endothelial eNOS/NAD(P)H-oxidase protein ratio in men with obesity.

### 3.3. Metabolic and Cardiovascular Outcomes

Six studies reported that physical activity, independent of weight loss, resulted in positive changes in whole body metabolic and cardiovascular outcomes and related biomarkers in people with obesity, including reductions in serum triglycerides and decreased arterial stiffness [23,27,28], increased mitochondrial respiration [25], increased fat oxidation and insulin sensitivity [19] and decreases in both liver fat and HbA1C [26]. Physical activity interventions in these studies included walking [23], high intensity [19,27] and moderate intensity cycling [26,27,28] as well as combined aerobic and resistance training [25].

In individuals with obesity who also have impaired glucose tolerance (IGT), McNeilly et al. [23] investigated the effects of a 12-week walking intervention on pulse wave velocity, blood pressure, fasting glucose, glycosylated hemoglobin, insulin, blood lipids and indices of oxidative stress and inflammation. Pulse wave velocity is a measure of arterial stiffness, as the existing literature shows that individuals with IGT tend to have greater arterial stiffness when compared to individuals without IGT. A total of 11 adults with obesity participated in 30 min walking exercise on a treadmill to reach 65% of their age-predicted maximum heart rate 5 times a week for 12 weeks and completed a journal for compliance. After completion of the intervention, investigators found a significant improvement in upper limb pulse wave velocity (9.08 + 1.27 m·s^−1^ vs. 8.39 + 1.21 m·s^−1^; *p* < 0.05) with a corresponding decrease in systolic blood pressure (*p* < 0.05). Subjects also experienced significant reductions in serum triglycerides ((1.52 + 0.53 mmol/L vs. 1.31 + 0.54 mmol/L) and a 34% decrease in lipid hydroperoxides compared to baseline (*p* < 0.05). No other metabolic biomarkers resulted in significant improvements. The limitations of this study included a small sample size and lack of a control group.

In order to understand how high intensity interval training (HIIT) or MICT affected endothelial dysfunction in individuals with obesity, a known precursor to atherosclerosis, Sawyer and colleagues [27] examined eighteen participants with a body mass index of 36.0 ± 5.0 kg/m^2^ over eight weeks. Brachial artery flow-mediated dilation and resting artery diameter were primary outcomes and are highly related to structural and functional adaptations that play a role in the improvements in vascular function. Brachial artery flow-mediated dilation increased after HIIT (5.13 ± 2.80% vs. 8.98 ± 2.86%, *p* = 0.02) and resting artery diameter increased after MICT (3.68 ± 0.58 mm vs. 3.86 ± 0.58 mm, *p* = 0.02). VO2 max increased (*p* < 0.01) similarly after HIIT (2.19 ± 0.65 L/min vs. 2.64 ± 0.88 L/min) and MICT (2.24 ± 0.48 L/min vs. 2.55 ± 0.61 L/min). Interestingly, HIIT required 27.5% less total exercise time and ∼25% less energy expenditure than MICT. This study was limited by a lack of dietary monitoring and the absence of a true sedentary control group.

Sabag et al. [26], examined the role of low-volume aerobic exercise on liver fat in 35 inactive adults with obesity living with type 2 diabetes. Subjects were randomized into one of the three groups by equally distributed, pregenerated lists of permuted blocks for a 12-week program of either HIIT, MICT, or placebo. Subjects that were in the MICT group exercised at 60% VO2 peak for 45 min, 3 days/week and subjects in the HIIT group exercised at 90% VO2 peak for 4 min, 3 days/week. After the intervention, researchers measured liver fat, HbA1C and cardiorespiratory fitness across all three groups. Researchers found liver fat decreased in both the MICT group (−0.9 +/− 0.7%) and HIIT group (1.7 +/− 1.1%), while increased in the placebo group (1.2 +/− 0.5%) (*p* = 0.046). HbA1C decreased in MICT (−0.3 +/− 0.03%) and HIIT (−0.3 +/− 0.03%) groups, while no improvement was observed in the placebo group (0.5 +/− 0.2%) (*p* = 0.014). Cardiorespiratory fitness also improved in both the MICT (2.3 +/− 1.2 mL/kg/min) and HIIT groups (1.1 +/− 0.5 mL/kg/min), but not in the placebo group (−1.5 +/− 0.9 mL/kg/min) (*p* = 0.006). Limitations of this study included a lack dietary monitoring and changes in energy expenditure outside of training as well as a small sample size.

In a secondary analysis of the data collected by Sabag and colleagues [26], Way et al. [28] evaluated the effect of exercise on cardiovascular health outcomes (central arterial stiffness and hemodynamic responses) in individuals with obesity and type 2 diabetes. With regard to arterial stiffness, there was a significant reduction in pulse wave velocity for both HIIT and MICT compared to placebo. Additional post-hoc analysis demonstrated no difference between the HIIT and MICT groups. There was a significant intervention effect for changes in VO2 peak (*p* = 0.01), glycosylated hemoglobin (*p* = 0.03), systolic blood pressure (*p* =0.01) and waist circumference (*p* = 0.03), which all improved following MICT or HIIT but not placebo; without differences between MICT and HIIT.

Mendham and colleagues [25] examined the role of exercise training on mitochondrial respiration and the association with altered intramuscular phospholipids in women with obesity. Over 12 weeks, 35 women were block randomized to an intervention or control group. The intervention group received supervised aerobic and resistance training at a moderate-vigorous intensity for 40–60 min, 4 days/week by a trained facilitator. After the 12 weeks, the intervention group was found to have a significant increase in mitochondrial respiration and content in response to exercise training. The metabolite and lipid signature at baseline were significantly associated with mitochondrial respiratory capacity (*p* < 0.05) but were not associated with whole-body insulin sensitivity or insulin regulated glucose transporter (GLUT4) protein content. Exercise training significantly altered the skeletal muscle lipid profile and these changes were associated with content-driven increases in mitochondrial respiration (*p* < 0.05), but not with the increase in whole-body insulin sensitivity or GLUT4 protein content.

The effects of SIT on substrate oxidation at rest and at submaximal exercise in people living with and without obesity were measured by Colpitts et al. [19]. Sixteen adults with obesity and eighteen adults without obesity took part in four weeks of interval training which consisted of sets of 30 s Wingate cycling with 4 min of active recovery three times a week with the number of sets increasing each week. At the completion of the intervention, researchers collected substrate oxidation estimations for resting state and during moderate-intensity exercise for both groups, as well as insulin sensitivity. Substrate oxidation was estimated using the respiratory exchange ratio from a resting metabolic rate test. Where appropriate, statistical analysis was corrected for fat free mass. While there was no change in weight or BMI in the experimental group, submaximal substrate oxidation improved significantly (reduction in RER) following four weeks of SIT in individuals living with obesity (baseline = 0.95 ± 0.08, post = 0.92 ± 0.06; *p* < 0.05) a change that was significantly different (*p* < 0.05) from individuals without obesity. Individuals living with obesity had an increase of 30.1% in insulin sensitivity, although this change did not reach statistical significance (baseline = 5.9 ± 3.4, post: 7.5 ± 7.0; *p* > 0.05). However, individuals without obesity had a lower, but significant, reduction in insulin sensitivity by 25.3% (baseline = 29.2 ± 18.4, post: 18.7 ± 14.2; *p* < 0.05) and significant differences were observed for absolute and percent changes in insulin sensitivity between groups (*p* < 0.05). Limitations of this study included a small sample size and no control group.

### 3.4. Systemic Outcomes

Three studies investigated whether physical activity improved systemic health, independent of weight loss, for those with obesity. Outcome markers included metabolic phenotype [20], measures of cardiorespiratory and muscle fitness [16] and quality of life with sustained exercise behavior change [22]. Physical activity modalities included community-based exercise classes [20], sports games [16] and resistance training [22].

Dalleck et al. [20] demonstrated that physical activity positively impacted the metabolic phenotype of individuals who have obesity. The possible benefits of community-based exercise training in transitioning metabolically abnormal obese (MAO) phenotypes to metabolically healthy but obese (MHO) phenotypes were observed. A total of 332 adults with a BMI > 30 kg/m^2^ participated in the study. Individuals who had 2–4 metabolic syndrome components were categorized as MAO. Those who had 1 or no component were considered MHO. Outcome variables assessed included absolute energy expenditure (EE), relative EE, waist circumference, BMI, body mass, systolic blood pressure (BP), diastolic BP, total cholesterol, HDL cholesterol, LDL cholesterol, triglycerides, glucose, cardiorespiratory fitness and 10-year risk score. Of these outcomes BMI, body mass, total cholesterol, LDL-cholesterol and 10-year risk score did not yield significant mean differences within the MAO to MHO group. After engaging in a community exercise program, 40% of metabolically abnormally obese (MAO) individuals transitioned to a metabolically healthy but obese (MHO) phenotype (*p* < 0.05). The major limitation of this retrospective study included not being able to control for the possibility that other factors (e.g., medication use and lifestyle, dietary and behavior changes) contributed to MAO to MHO phenotype transition, irrespective of the community exercise program.

With regard to markers of cardiorespiratory health and muscle strength, Biddle and colleagues [16] reported improvement in cardiorespiratory fitness and leg strength in Pacific adults who participated in informal sports. Subjects included 20 adults (13 females, 7 males) engaging in 45 min small-sided games of soccer, basketball, volleyball, touch rugby, cricket and other non-conventional games such as ‘chain tag,’ ‘rob the nest’ and ‘bullrush.’ Participants were randomized in a computer-generated 1:1 fashion but were not blinded. The control group was asked not to change their physical activity routine for the next 4 weeks. Primary outcomes of these small-sided sessions included cardiorespiratory fitness (VO2 peak) and leg strength (maximal concentric force of quadriceps at 60 degree/s). Outcomes were both measured at baseline and at 4 weeks. Secondary outcomes included glycemia (fasting glucose, HbA1c), lipid profile (total cholesterol, HDL, triglycerides), blood pressure (BP) and inflammatory markers (C-reactive protein). Results demonstrated that changes in outcomes were greater within the intervention group than in the control group (VO2 peak: 0.9 L/min (*p* = 0.003), leg strength: 17.8 N.m (*p* = 0.04) and HDL: 0.12 mmol/L (*p* = 0.02)). Limitations of this study included a small sample size, unblinded subject pool and a short duration.

Physical activity in the form of resistance training was also found to improve muscular strength and quality of life for adults with obesity and type 2 diabetes [22]. Over 16 weeks, 48 individuals with obesity and type 2 diabetes were assigned to either resistance training (RT) (*n* = 27) or the control group (*n* = 21). There were no statistically significant differences between the groups at baseline for any of the study variables. RT included access to a multi-gym with dumbbells and home supervision from a certified personal trainer. Primary outcome was muscular strength (33% increase, *p* < 0.001) and RT behavior. To assess muscular strength, participants performed one rep max (1RM) tests of seated chest press, seated row and leg press. Chest press, seated row and leg press 1RMs were combined to create a total strength score. The resistance training program was found to have a significant effect on resistance training behavior. During the 16 weeks, participants reported the number and duration of each resistance training session completed. Results showed a significant positive association between changes in RT behavior (C = 110.99, 95%, CI = 78.28 to 143.72, *p* < 0.001) and changes in RT planning strategies (r = 0.51, *p* < 0.01) when controlling for the intervention effect. Limitations of this study included a small sample size and self-reported behavioral data.

### 3.5. Brain Health Outcomes

Two studies investigated whether physical activity improved brain health markers, independent of weight loss, for those with obesity. Outcomes included changes in sleep [24], depressive symptoms [21,25], quality of life and emotional health [21]. Physical activity modalities included combined aerobic and resistance training [24] and tai chi [21].

Mendham et al. [24] evaluated whether exercise training had an impact on sleep quality and depressive symptoms in a previously reported cohort [25] of women with obesity living in low socioeconomic communities. Participants were block randomized to exercise or control groups and no differences were noted between groups at baseline except in VO2 peak. The exercise group completed 12 weeks of combined resistance and aerobic training (40–60 min, 4 d/wk) and the control group maintained habitual diet and activity. The effect of exercise training intervention on sleep and depression, sleep characteristics, peak oxygen consumption and glucose metabolism were assessed over 12 weeks. This intervention indicated improvement in sleep quality (*p* < 0.001), sleep efficiency (*p* = 0.005) and depressive symptoms (*p* = 0.002). These results were further associated with other components such as improved peak oxygen consumption and sedentary time. Depressive symptoms improved with peak oxygen consumption (*p* < 0.001) while sleep improvement was correlated with reduced sedentary time (*p* = 0.018). This proof-of-concept study was limited by a small sample size, multiple statistical testing and subjectively measured sleep quality.

Liu et al. [21] evaluated the effect of tai chi on quality of life in adults with central obesity and depression. Over 24 weeks, 213 participants were randomly assigned to either a tai chi group or usual medical care, with no differences noted between groups at baseline. Those engaging in the tai chi group participated in 3 × 1.5 h sessions per week under supervision. Quality of life was assessed prior to the intervention, 12 weeks and post-intervention at 24 weeks using the Medical Outcome Study (MOS) SF-36 survey. Outcome measures included general health, physical functioning, role in physical health, role in emotional health, social functioning, bodily pain, mental health and vitality. Subjects were found to have moderately severe levels of depression at baseline. Results depicted significant improvement within the tai chi group in three of the SF-36 subscale scores—physical functioning (*p* < 0.01), role in physical health (*p* < 0.01 and role in emotional health (*p* < 0.01). Additionally, the following outcomes also demonstrated significant interaction: physical functioning (*p* < 0.01), role-physical (*p* < 0.01), bodily pain (*p* < 0.01) and role-emotional (*p* < 0.01), but not in general health, vitality, social functioning and mental health. Limitations of this study included not categorizing patients by specific clinical categorization of depression.

## 4. Discussion

Physical activity has beneficial physiologic effects from the cellular to systemic level, independent of weight loss, in individuals with obesity. Yet, outcomes have been severely under-studied in this growing population. The current systematic review found only 14 interventional studies of people with obesity that examined health outcomes related to physical activity that were not primarily dependent on weight loss. This is despite the current prevalence of obesity in the United States of 42.4% [1], with projections to reach over 50% of the population by 2030 and severe obesity likely to become the most common BMI category among women, non-Hispanic black adults and low-income adults [30]. This novel systematic review highlights how a lack of data on the effects of physical activity for non-weight-related physical and mental health outcomes in individuals with obesity is a critical missed opportunity for the obesity research community, with implications for healthcare counseling strategies and public policy initiatives.

Positive outcomes at the cellular level indicate potential benefit in risk reduction for related comorbidities of obesity and include alterations in telomere length [17], improvement in anti-oxidative and anti-thrombotic enzymes associated with HDL cholesterol [29] and skeletal muscle microvascular enzymes that affect capillary density and vasoreactivity [18]. From a metabolic standpoint, increased physical activity leads to reductions in serum triglycerides and decreased arterial stiffness [23,27,28], increased mitochondrial respiration efficiencies [25], increased fat oxidation and insulin sensitivity [19] and decreases in both liver fat and HbA1C [26]. Together, these physiologic markers represent a possible reduction in the development of cardiovascular disease and type 2 diabetes in people that have obesity, leading to reduced risk of mortality and related healthcare costs [3,4,31,32].

When examining the systemic benefits of physical activity for individuals with obesity, outcome markers included alterations in metabolic phenotype [20], improved measures of cardiorespiratory and muscle fitness [16], enhanced quality of life and sustained exercise behavior [22]. Moreover, physical activity has specific benefits for brain health including improvements in sleep [24] and depressive symptoms [21,24], as well emotional health and overall quality of life [21]. The systemic and brain health benefits of physical activity are indeed highlighted in the Physical Activity Guidelines for Americans, yet little to no information was provided about the influence, if any, of weight status on the relationship between physical activity and measures of depressive symptoms or sleep [33].

Of particular note for the research community is the lack of long-term interventional studies examining consistent and repeatable physical activity modalities that investigate the benefits of physical activity in people with obesity, independent of weight loss. In the current systematic review, studies ranged from 4–16 weeks. Modalities included combination aerobic and resistance training, walking, resistance training alone, cycling, high intensity interval training and tai chi. Conducting extended interventional studies, with consistent and comparable modalities of physical activity, would provide significant opportunities to understand the benefits of physical activity in all individuals with obesity and also between obesity subclasses [14].

The research gap has implications for how healthcare providers are educated on the benefits of physical activity for individuals with obesity and related counseling strategies and practices. It is critical that data, education and clinical practice patterns be linked in order to affect behavior change in patients. A change in education and counseling strategies, driven by a wider research base, could have critical implications for initiating and sustaining physical activity behaviors in patients with obesity. Indeed, calls have been made to assess physical activity and cardiorespiratory fitness as a vital sign in clinical settings [34], a practice which could be particularly beneficial for high-risk populations [35]. Yet, cardiorespiratory fitness is not mentioned in the current guidelines for the management of obesity [36,37] and physical activity is only recommended to the people with obesity as a tool to induce negative energy balance [1,2]. This highlights both a dilemma and also an opportunity for education and improvement in healthcare professional training [38].

Current communication strategies and skills surrounding physical activity recommendations are known to be lacking and misaligned with behavior change [39,40]. Too narrow a focus—although intended to improve health through weight loss—may instead contribute to the high prevalence of weight gain and weight cycling [41], creating a “weight loss futile cycle” [3], which can also be associated with significant health risks. Weight loss by itself is not a behavior and by not examining other outcomes that could make physical activity more relevant and compelling for patients to initiate and sustain [39], the messaging may be causing more harm than good.

The limitations of this review include examining only studies with subjects with a BMI over 30 kg/m^2^. It is likely the noted benefits are generalizable across subsets of BMI. However, this strict inclusion criteria could also be viewed as a strength as it fills a gap in the literature. Limitations could also include a restriction to interventional study designs, potentially missing beneficial correlational data.

## 5. Conclusions

The current systematic review demonstrates that for individuals with obesity, there are both small-scale and large-scale physiologic benefits that occur with increased physical activity of various modalities. The positive health outcomes of these changes need to be emphasized to individuals with obesity more than just the general recommendation of weight loss. This will likely require the coordinated efforts of healthcare providers and physical activity practitioners as they work to improve the health of individuals with a BMI over 30. However, as research on best practices is still lacking, more long-term studies that include a variety of stakeholders to understand how to inform and encourage sustainable physical activity behaviors in individuals with obesity are needed. Linking innovative research with education and clinical practice patterns has critical implications for behavior change and overall health in individuals with obesity.

## Figures and Tables

**Figure 1 ijerph-19-04981-f001:**
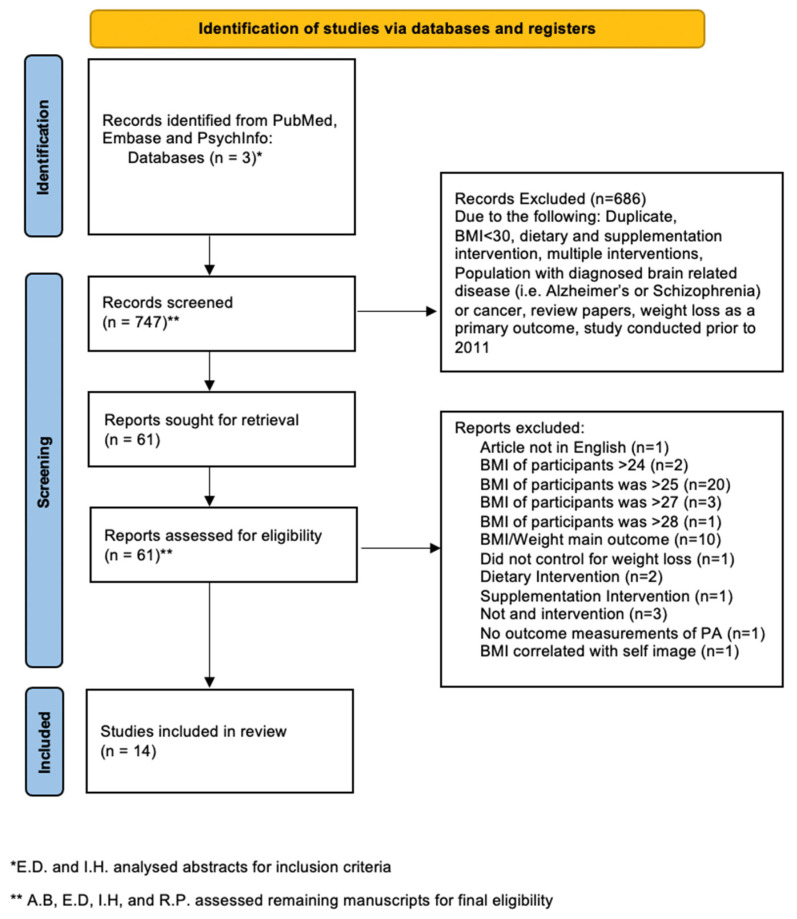
Flowchart depicting the choice of studies.

**Table 1 ijerph-19-04981-t001:** All search terms.

Database	Search Terms Used
Embase	intervention AND obes# AND ‘physical activity’ AND fitness AND physical AND exercise NOT ‘body weight loss’ NOT (weight AND reduction) NOT ‘weight management’ NOT (lose AND weight) NOT (body AND composition AND assessment) AND [2011–2021]/py AND ([adult]/lim OR [aged]/lim)
PsychInfo	Intervention AND obes* AND physical activity or exercise or fitness or physical exercise NOT weight loss NOT weight reduction NOT weight management NOT lose weight NOT body composition assessment; Adults 18–64, 65+; PY 2011–2021
Pubmed	(body mass index OR “obes *” or “obesity paradox” OR “obesity treatment”) AND (“physical activity” or “exercise” or “fitness”) AND (reducing health risks) AND (weight neutral)
Pubmed	(body mass index) AND (humans) AND (life style) AND (“adult *”) OR (“obes *”) AND (“physical activity” or “exercise”) AND (health benefits independent of weight status)
Pubmed	(body mass index OR “obes *” or “obesity paradox”) AND (“physical activity” or “exercise”) AND (Cardiorespiratory fitness) AND (“fitness and fatness”)
Pubmed	(adult) OR (aged) OR (middle aged) AND (body mass index) AND (humans) AND (life style) AND (mortality / trends) OR (obesity/mortality) OR (overweight/mortality) AND (united states/epidemiology) AND (risk reduction behavior)
Pubmed	(body mass index OR “obes *” or “obesity paradox” OR “obesity treatment”) AND (“physical activity” or “exercise” or “fitness”) AND (“CRF”) AND (“fitness and fatness”) AND (reducing health risks)

**Table 2 ijerph-19-04981-t002:** Included articles.

Author	Year Published	Country	Outcome Category	Physical Activity Modality	Subjects ^a^	Intervention Duration and Frequency	Primary Outcome	Results	BMI (kg/m^2^)
Biddle, M.G. et al. [16]	2011	New Zealand	Systemic	Sports Games	Sedentary Pacific Adults from age 16–65; *n* = 9 INT, *n* = 11 CON	45 min per day, 3 times per week, for 4 weeks	Cardiorespiratory fitness and leg strength	Increase in VO2 peak (*p* = 0.003), leg strength (*p* = 0.04) and HDL (*p* = 0.02) were greater within the intervention versus control group	36.3
Brandao, C.F.C. et al. [17]	2020	Brazil	Cellular	Combined Aerobic and Resistance Training	Sedentary Females from age 20–40; *n* = 20 (No CON)	55 min per day, 3 times per week, for 8 weeks	Telomere length	Increased telomere length, fat-free mass and VO2 max. Inverse relationship between telomere length and waist circumference (*p* < 0.05)	34
Cocks, M. et al. [18]	2016	United Kingdom	Cellular	SIT or MICT using a cycle ergometer	Sedentary Young Men; *n* = 8 INT; *n* = 8 CON	SIT: 4–7 constant workload intervals of 200% Wmax 3 times per week; MICT: 40–60 min cycling at 65% VO2 peak, 5 times per week, for 4 weeks	Skeletal muscle microvascular density and microvascular filtration capacity	SIT and MICT have equal benefits on aerobic capacity, insulin sensitivity, muscle capillarization and endothelial protein ratio men with obesity (*p* < 0.05)	34.8
Colpitts, B.H. et al. [19]	2021	Canada	Metabolic and Cardiovascular	SIT using a cycle ergometer	Inactive adults between age 16–60 years; *n* = 12 with obesity, *n* = 18 without obesity	30 s Wingate cycling with 4 min of active recovery, 3 times per week, for 4 weeks	Substrate oxidation at rest and at submaximal exercise	Significant increase in fat oxidation during exercise for adults living obesity (*p* = 0.001)	34.1
Dalleck, L.C. et al. [20]	2014	USA	Systemic	Supervised community-based exercise program	Healthy Adults aged 22–88 years; *n* = 55 MAO, *n* = 37 MHO, *n* = 200 MH not obese; *n* = 24 MA not obese	Personalized time periods, 3 times per week, for 14 weeks	Metabolic Phenotype	40% of metabolically abnormally obese (MAO) individuals transitioned to a metabolically healthy but obese (MHO) phenotype (*p* < 0.05)	34.1
Liu, Xin et al. [21]	2019	Australia	Brain Health	Tai Chi	Adults with a diagnosis of depression; *n* = 106 INT *n* = 107 CON	1.5 h sessions, 3 times per week, for 24 weeks	Quality of life and emotional health	Improvement of physical functioning, role in physical health and role in emotional health (*p* < 0.05)	>30
Lubans, D.R. et al. [22]	2012	Canada	Systemic	Supervised Resistance Training	Sedentary adults with Type 2 Diabetes; *n* = 27 INT; *n* = 21 CON	3 times per week, for 16 weeks	Muscular strength and resistance training behavior	Increased muscle strength and resistance training behavior (*p* < 0.05)	36
McNeilly, A.M. et al. [23]	2012	United Kingdom	Metabolic and Cardiovascular	Treadmill Walking	Adults with impaired glucose tolerance; *n* = 11 (no CON)	30 min walking at 65% maximum heart rate, 5 times a week, for 12 weeks	Arterial stiffness	Improvement in Pulse wave velocity, systolic blood pressure, triglycerides and lipid hydroperoxides (*p* < 0.05)	32.4
Mendham, A.E. et al. [24]	2021	South Africa	Brain Health	Combined Aerobic and Resistance Training	Low Income adults women aged 20–35 years; *n* = 20 INT; *n* = 15 CON	40–60 min at a moderate-vigorous intensity, 4 days per week for 12 weeks	Depressive Symptoms and Sleep quality	Depressive symptoms improved with peak oxygen consumption (*p* < 0.001) while sleep improvement was correlated with reduced sedentary time (*p* = 0.018).	30–40
Mendham, A.E. et al. [25]	2021	South Africa	Metabolic and Cardiovascular	Combined Aerobic and Resistance Training	Low Income Black adult women aged 20–35 years; *n* = 20 INT; *n* = 15 CON	40–60 min at a moderate-vigorous intensity, 4 days per week for 12 weeks	Mitochondrial respiration and the association with altered intramuscular phospholipid signature in women with obesity	Exercise training significantly altered the skeletal muscle lipid profile and increased content driven mitochondrial metabolism (*p* < 0.05)	30–40
Sabag, A. et al. [26]	2020	Australia	Metabolic and Cardiovascular	HIIT training/cycling MICT training/cycling or CON	Inactive Adults with Type 2 Diabetes; *n* = 12 MICT; *n* = 12 HIIT; *n* = 11 CON	HIIT: 19 total minutes of intervals designed to elicit 90% VO2 peak; MICT: 45 min at 60% VO2 peak, 3 days per week, for 12 weeks	Liver fat, glucose metabolism, cardiorespiratory fitness	Decreased liver fat (*p* = 0.046), HbA1C (*p* = 0.014) and improved cardiorespiratory fitness (*p* = 0.006) in MICT/HIIT but not placebo	35.9
Sawyer B.J et al. [27]	2016	USA	Metabolic and Cardiovascular	HIIT training/cycling MICT training/cycling	Healthy adults aged 18–55 years; *n* = 11 HIIT; *n* = 11 MICT	HIIT: 10 × 1 min intervals at 90–95% maximum heart rate; MICT: 30 min at 70–75% maximum heart rate, 3 sessions per week, for 8 weeks (3 sessions/wk) of either	Endothelial function and maximum oxygen uptake	HIIT improved brachial artery flow-mediated dilation (*p* = 0.02) and MICT increased resting artery diameter (*p* = 0.02).HIIT and MICT both enhanced low flow-mediated constriction (*p* = 0.03) and VO2 max in both groups (*p* < 0.01)	
Way, K.L. et al. [28]	2020	Australia	Systemic	HIIT training/cycling MICT training/cycling or CON	Inactive Adults with Type 2 Diabetes; *n* = 12 MICT; *n* = 12 HIIT; *n* = 11 CON	HIIT: 19 total minutes of intervals designed to elicit 90% VO2 peak; MICT: 45 min at 60% VO2 peak, 3 days per week, for 12 weeks	Central arterial stiffness and hemodynamic responses	Significant reduction in pulse wave velocity (*p* = 0.03) HbA1c (*p* = 0.03), systolic blood pressure (*p* = 0.01) and waist circumference (*p* = 0.03),	36.1
Woudberg N.J. et al. [29]	2018	South Africa	Metabolic and Cardiovascular	Combined Aerobic and Resistance Training	Low Income Black adult women aged 20–35 years; *n* = 20 INT; *n* = 15 CON	40–60 min at a moderate-vigorous intensity, 4 days per week for 12 weeks	Lipid profile and HDL functionality	Decreased BMI (*p* = 0.01), increased antioxidant capacity (*p* = 0.02) and anti-thrombotic function (*p* = 0.002)	30–40

^a^ Abbreviations: CON—Control; INT—Intervention; MICT—Moderate Intensity Continuous Training; HIIT—High Intensity Interval Training; SIT—Sprint Interval Training; MAO—metabolically abnormally obese; MHO—metabolically healthy but obese (MHO); HDL—High Density Lipoprotein.

## Data Availability

Not applicable.

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
