# Peer review of "The Benefits of Physical Activity for People with Obesity, Independent of Weight Loss: A Systematic Review"

_ijerph, 2022, doi:10.3390/ijerph19094981_

Round 1
Reviewer 1 Report
This is an interesting review of the literature on RCTs with an intervention of physical activity without the intention of weight loss, to examine physiologic changes in an obese population.
Introduction
The first two paragraphs are off-topic because they are dedicated to weight loss. While it is important to discuss increasing obesity in the population, given the research question, it would be better to focus on the health risks associated with obesity rather than the failure of weight loss programs.
Line 59 - "...perhaps leading to incorrect course of treatment" is vague. Do you mean that people are treated for the problems of physical and mental health when these might improve with physical exercise?
Methods
Please confirm that all studies are RCT (as stated in line 60). It looks like some studies may be case-control study design. Do they all have an intervention and a control group; are any of them cohort studies that measure change in the individual before and after the intervention?
Results
The text in the results section would be better presented in a table.
Table 2 should describe the studies included in the review (Author, year, country, population, number of participants in intervention and control groups, study design).
Add Table 3 for summary of results from the studies (Author, intervention, control, outcome of interest, outcome intervention group, outcome control group, difference between the group). This table can be separated in the 4 sections of Cellular, Metabolism, Systemic, Brain Health
Have the studies controlled for potential confounding; age, sex, comorbidities, medication, weight/BMI etc?
Line 165 - Impaired glucose intolerance should read impaired glucose tolerance.
Discussion
The first two paragraphs of the discussion would be better in the Introduction.
Author Response
Reviewer #1
This is an interesting review of the literature on RCTs with an intervention of physical activity without the intention of weight loss, to examine physiologic changes in an obese population.
Thank you, we appreciate your thorough and kind review and have addressed your concerns below.
Introduction
The first two paragraphs are off-topic because they are dedicated to weight loss. While it is important to discuss increasing obesity in the population, given the research question, it would be better to focus on the health risks associated with obesity rather than the failure of weight loss programs.
We have shortened the topic of weight loss in the discussion and, as per your comment regarding the first two paragraphs of the discussion, have moved that content into the introduction. Lines 33-43
Line 59 - "...perhaps leading to incorrect course of treatment" is vague. Do you mean that people are treated for the problems of physical and mental health when these might improve with physical exercise?
This has been edited for clarity.
Methods
Please confirm that all studies are RCT (as stated in line 60). It looks like some studies may be case-control study design. Do they all have an intervention and a control group; are any of them cohort studies that measure change in the individual before and after the intervention?
Results
The text in the results section would be better presented in a table.
We attempted to add this text to a table, but it became impossible to format according to the journals’ instructions to authors due to wordiness. As such, we have addressed this issue by adding specific columns requested below in our table and shortening the text in the results section.
Table 2 should describe the studies included in the review (Author, year, country, population, number of participants in intervention and control groups, study design).
The columns of country, subjects (including n=INT and n=CON) and study design have been added into the table.
Add Table 3 for summary of results from the studies (). This table can be separated in the 4 sections of Cellular, Metabolism, Systemic, Brain Health
For brevity and formatting issues, we kept all data in one table. It has been separated by section.
Have the studies controlled for potential confounding; age, sex, comorbidities, medication, weight/BMI etc?
Upon further investigation, it seems that none of the studies controlled for any of the above mentioned potential confounders. Many stated that their small study size may have reduced the effects of confounding effects. One study in general, Sabag et al, stated that they did not find a between group difference in body weight change which they concluded means that the confounder may be negligible or affect each group equally.
Line 165 - Impaired glucose intolerance should read impaired glucose tolerance.
This has been fixed.
Discussion
The first two paragraphs of the discussion would be better in the Introduction.
We have shortened the topic of weight loss in the discussion and, as per your comment regarding the first two paragraphs of the discussion, have moved that content into the introduction.
Reviewer 2 Report
The analysis of the literature data on the benefits of physical activity for obese people highlights very important aspects of this growing phenomenon, both among adults and children. As the authors themselves emphasize, the research requires continuation and extension. Currently, I do not have any major comments to the reviewed article, which can be printed as it is.
The analysis of the literature data on the benefits of physical activity for obese people highlights very important aspects of this growing phenomenon, both among adults and children. As the authors themselves emphasize, the research requires continuation and extension.
Strength of this manuscripts is the lack of information in the related literature, this is an interesting review of the literature on RCT
Minor revision, weakness
Introduction
Move the first two paragraphs of the Discussion to Introduction.
Line 50-52 please write the references.
Methods
Confirm that all studies have a control group and are RTC
Results
Table 2 should describe the studies included in the review
Author Response
Reviewer #2
The analysis of the literature data on the benefits of physical activity for obese people highlights very important aspects of this growing phenomenon, both among adults and children. As the authors themselves emphasize, the research requires continuation and extension. Currently, I do not have any major comments to the reviewed article, which can be printed as it is.
The analysis of the literature data on the benefits of physical activity for obese people highlights very important aspects of this growing phenomenon, both among adults and children. As the authors themselves emphasize, the research requires continuation and extension.
Strength of this manuscripts is the lack of information in the related literature, this is an interesting review of the literature on RCT
Minor revision, weakness
Thank you so much for your careful and thorough review of this manuscript and have addressed the concerns presented in your comments
Introduction
Move the first two paragraphs of the Discussion to Introduction.
We have shortened the topic of weight loss in the discussion and, as per your comment regarding the first two paragraphs of the discussion, have moved that content into the introduction. Page 1, paragraph 1 & 2.
Line 50-52 please write the references.
The references have been added as requested.
Methods
Confirm that all studies have a control group and are RTC
We have confirmed that 11 out of the 13 studies included RTCs with control groups. The other two studies were prescribed the same exercise to all participants. We updated this in the abstract, introduction (Line 62) and in the Tables.
Results
Table 2 should describe the studies included in the review
We have added several columns to help further describe the studies including country, subjects (including n=INT and n=CON) and study design.
Reviewer 3 Report
This systematic review by Pojednic et al investigates randomized controlled trials that examine health markers, independent of weight loss, in obese individuals after different types of physical activity. This is a very well organized, logically written, timely and interesting review. Among the strengths of this manuscripts is this it feels the lack of information in the related literature, since most of the studies concentrated on beneficial effect of exercises of weight loss in obese individuals. Authors identified fourteen studies that utilized a variety of physical activity interventions with 4 primary endpoints that included cellular, metabolic, systemic, and brain health outcomes. Modalities included combination aerobic and resistance training, walking, resistance training alone, cycling, high intensity interval training, and tai chi. The thorough review of the literature demonstrates that for individuals with obesity, there are both small-scale and large-scale physiologic benefits that occur with increased physical activity of various modalities. Authors concluded that focusing on the benefits, rather than a narrow focus of weight loss alone, may increase physical activity behavior and health for individuals with obesity. I do not identify major weaknesses. There are following minor weaknesses: authors need to indicate the novelty and limitations and of the study; to highlight the efforts to improve the general health.
- Authors need to provide imitations of this review in order to provide a context for the results.
1. Authors need to indicate the novelty and limitations of this review in order to provide a context for the results.
2. Authors need to highlight the efforts that research/clinic and physical activity services need to cooperate to improve the general health.
3. Lane 52: “There are limited intervention studies …”authors need to provide the references.
Author Response
This systematic review by Pojednic et al investigates randomized controlled trials that examine health markers, independent of weight loss, in obese individuals after different types of physical activity. This is a very well organized, logically written, timely and interesting review. Among the strengths of this manuscripts is this it feels the lack of information in the related literature, since most of the studies concentrated on beneficial effect of exercises of weight loss in obese individuals. Authors identified fourteen studies that utilized a variety of physical activity interventions with 4 primary endpoints that included cellular, metabolic, systemic, and brain health outcomes. Modalities included combination aerobic and resistance training, walking, resistance training alone, cycling, high intensity interval training, and tai chi. The thorough review of the literature demonstrates that for individuals with obesity, there are both small-scale and large-scale physiologic benefits that occur with increased physical activity of various modalities. Authors concluded that focusing on the benefits, rather than a narrow focus of weight loss alone, may increase physical activity behavior and health for individuals with obesity. I do not identify major weaknesses. There are following minor weaknesses: authors need to indicate the novelty and limitations and of the study; to highlight the efforts to improve the general health.
Thank you for this thorough review and comments. We have addressed the noted issues below.
- Authors need to indicate the novelty and limitations of this review in order to provide a context for the results.
The novelty of this systematic review have been added to the first paragraph of the discussion (Lines 337-340). Limitations have been addressed in a new paragraph in the discussion (Lines398-402).
- Authors need to highlight the efforts that research/clinic and physical activity services need to cooperate to improve the general health.
A sentence has been added to the final paragraph of the discussion that addresses the need for collaborative efforts within and between sectors to encourage and sustain physical activity behaviors (Lines409-413).
- Lane 52: “There are limited intervention studies …”authors need to provide the references.
A reference has been added to support this statement.
Reviewer 4 Report
The authors do a thorough systematic review of the current published data on the effects of physical activity for individuals with obesity independent of body weight loss. the authors selected a surprisingly good enough number of studies to review the effects of physical activity on several weight-independent health factors. As the authors stated, more studies and evaluations of long physical activity interventions are desired. The information presented and the aim of the review are highly relevant and of interest for the public health strategies to treat obesity.
Minor comments:
After the second paragraph in the introductions 9line 39) it would be nice to add a phrase or 2 about the benefits of exercise in different health parameters or pleiotropic effects of physical activity that accompany loss, as an introduction of these beneficial effects independently of weight loss.
Line 59-61. to examine randomized controlled trials that examine the effects of physical activity in health markers, …
Line 68-69. Change structure of the sentence: A systematic review method that complied with the Preferred Reporting Items for Systematic Review and Meta-Analysis Protocols (PRISMA-P) was used [18]
In section 3.3 (line 157). I suggest dividing the section into 2 subcategories: metabolic outcomes and cardiovascular health outcomes. or changing the group section titles for Metabolic and cardiovascular health outcomes
Move study 25 to last category (Brain outcomes): quality of life with sustained exercise behavior change[25]
Line 310. give a little more details on how the study that is being cited assed quality of life prior to the intervention.
Author Response
The authors do a thorough systematic review of the current published data on the effects of physical activity for individuals with obesity independent of body weight loss. The authors selected a surprisingly good enough number of studies to review the effects of physical activity on several weight-independent health factors. As the authors stated, more studies and evaluations of long physical activity interventions are desired. The information presented and the aim of the review are highly relevant and of interest for the public health strategies to treat obesity.
Thank you for the positive feedback and your rigorous review of the manuscript. We appreciate your assessment of the review’s potential for impacting public health strategies. We have given your comments full consideration and have addressed them as noted below.
Minor comments:
After the second paragraph in the introductions 9line 39) it would be nice to add a phrase or 2 about the benefits of exercise in different health parameters or pleiotropic effects of physical activity that accompany loss, as an introduction of these beneficial effects independently of weight loss.
Language addressing the pleiotropic effects of physical activity has been added (Lines 33-42).
Line 59-61. to examine randomized controlled trials that examine the effects of physical activity in health markers, …
This language has been added (Lines 62-63)
Line 68-69. Change structure of the sentence: A systematic review method that complied with the Preferred Reporting Items for Systematic Review and Meta-Analysis Protocols (PRISMA-P) was used [18]
This sentence structure has been changed (Line 72).
In section 3.3 (line 157). I suggest dividing the section into 2 subcategories: metabolic outcomes and cardiovascular health outcomes. or changing the group section titles for Metabolic and cardiovascular health outcomes
The title has been adjusted to read Metabolic and Cardiovascular Outcomes.
Move study 25 to last category (Brain outcomes): quality of life with sustained exercise behavior change[25]
We carefully considered this revision and determined this study should remain in the systemic category. The primary outcome was muscular strength. Quality of life with sustained exercise behavior change was a secondary outcome marker for systemic and is distinct from quality of life for brain health. Edits have been made to improve clarity of primary outcomes (Lines 285-286).
Line 310. give a little more detail on how the study that is being cited assessed quality of life prior to the intervention.
This has been edited for clarity.
Round 2
Reviewer 1 Report
The content of sentence starting on line 39 is repeated in the following paragraph.
The authors could expand on the methods/results by describing how the studies were reviewed for quality and limitations specific to each of the included studies. For example, were the intervention and control arms matched on age, sex, or other criteria?
There are some errors in the age ranges and inconsistencies in the format of the text included under "Subject" in Table 2. It would be clearer if "INT" and "CON" were replaced with "Intervention" and "control", or these were defined in the footnote.
Inconsistencies in terms and acronyms used throughout. For example, type 2 diabetes is referred to as type 2 diabetes and T2D (line 199). Similarly, acronyms are used and redefined throughout.
Author Response
Thank you so much for your continuing thorough review of our manuscript. We have made updates and corrections to our work as outlined below.
The content of sentence starting on line 39 is repeated in the following paragraph.
These two paragraphs have been combined and restructured to avoid redundancy (lines 39-52).
The authors could expand on the methods/results by describing how the studies were reviewed for quality and limitations specific to each of the included studies. For example, were the intervention and control arms matched on age, sex, or other criteria?
Where available, randomization, statistical controls, and limitations were included for each study. A sentence has been added to the inclusion criteria to address limitations noted by authors.
There are some errors in the age ranges and inconsistencies in the format of the text included under "Subject" in Table 2. It would be clearer if "INT" and "CON" were replaced with "Intervention" and "control", or these were defined in the footnote.
Ages have been checked and corrected. A footnote has been added to clarify all abbreviations.
Inconsistencies in terms and acronyms used throughout. For example, type 2 diabetes is referred to as type 2 diabetes and T2D (line 199). Similarly, acronyms are used and redefined throughout.
All acronyms have been checked and updated.